# Integrative Analysis of Next-Generation Sequencing for Next-Generation Cancer Research toward Artificial Intelligence

**DOI:** 10.3390/cancers13133148

**Published:** 2021-06-24

**Authors:** Youngjun Park, Dominik Heider, Anne-Christin Hauschild

**Affiliations:** 1Department of Mathematics and Computer Science, Philipps-University of Marburg, 35032 Marburg, Germany; youngjun.park@uni-marburg.de (Y.P.); dominik.heider@uni-marburg.de (D.H.); 2Department of Medical Informatics, University Medical Center Göttingen, 37075 Göttingen, Germany

**Keywords:** next-generation sequencing, systems biology, pathway, biological network, machine learning, deep neural network, artificial intelligence

## Abstract

**Simple Summary:**

In recent years both research areas of next-generation sequencing and artificial intelligence have grown remarkably. Their intersection simultaneously gave rise to a panacea of different algorithms and applications. This article delineates tailored machine learning and systems biology approaches and combinations thereof that tackle the various challenges that arise in the face of big data. Moreover, it provides an overview of the numerous applications of artificial intelligence aiding the analysis and interpretation of next-generation sequencing data.

**Abstract:**

The rapid improvement of next-generation sequencing (NGS) technologies and their application in large-scale cohorts in cancer research led to common challenges of big data. It opened a new research area incorporating systems biology and machine learning. As large-scale NGS data accumulated, sophisticated data analysis methods became indispensable. In addition, NGS data have been integrated with systems biology to build better predictive models to determine the characteristics of tumors and tumor subtypes. Therefore, various machine learning algorithms were introduced to identify underlying biological mechanisms. In this work, we review novel technologies developed for NGS data analysis, and we describe how these computational methodologies integrate systems biology and omics data. Subsequently, we discuss how deep neural networks outperform other approaches, the potential of graph neural networks (GNN) in systems biology, and the limitations in NGS biomedical research. To reflect on the various challenges and corresponding computational solutions, we will discuss the following three topics: (i) molecular characteristics, (ii) tumor heterogeneity, and (iii) drug discovery. We conclude that machine learning and network-based approaches can add valuable insights and build highly accurate models. However, a well-informed choice of learning algorithm and biological network information is crucial for the success of each specific research question.

## 1. Introduction

The development and widespread use of high-throughput technologies founded the era of big data in biology and medicine. In particular, it led to an accumulation of large-scale data sets that opened a vast amount of possible applications for data-driven methodologies. In cancer, these applications range from fundamental research to clinical applications: molecular characteristics of tumors, tumor heterogeneity, drug discovery and potential treatments strategy. Therefore, data-driven bioinformatics research areas have tailored data mining technologies such as systems biology, machine learning, and deep learning, elaborated in this review paper (see Figure 1 and Figure 2). For example, in systems biology, data-driven approaches are applied to identify vital signaling pathways [1]. This pathway-centric analysis is particularly crucial in cancer research to understand the characteristics and heterogeneity of the tumor and tumor subtypes. Consequently, this high-throughput data-based analysis enables us to explore characteristics of cancers with a systems biology and a systems medicine point of view [2].Combining high-throughput techniques, especially next-generation sequencing (NGS), with appropriate analytical tools has allowed researchers to gain a deeper systematic understanding of cancer at various biological levels, most importantly genomics, transcriptomics, and epigenetics [3,4]. Furthermore, more sophisticated analysis tools based on computational modeling are introduced to decipher underlying molecular mechanisms in various cancer types. The increasing size and complexity of the data required the adaptation of bioinformatics processing pipelines for higher efficiency and sophisticated data mining methodologies, particularly for large-scale, NGS datasets [5]. Nowadays, more and more NGS studies integrate a systems biology approach and combine sequencing data with other types of information, for instance, protein family information, pathway, or protein–protein interaction (PPI) networks, in an integrative analysis. Experimentally validated knowledge in systems biology may enhance analysis models and guides them to uncover novel findings. Such integrated analyses have been useful to extract essential information from high-dimensional NGS data [6,7]. In order to deal with the increasing size and complexity, the application of machine learning, and specifically deep learning methodologies, have become state-of-the-art in NGS data analysis.

Therefore, a large number of studies integrate NGS data with machine learning and propose a novel data-driven methodology in systems biology [8]. In particular, many network-based machine learning models have been developed to analyze cancer data and help to understand novel mechanisms in cancer development [9,10]. Moreover, deep neural networks (DNN) applied for large-scale data analysis improved the accuracy of computational models for mutation prediction [11,12], molecular subtyping [13,14], and drug repurposing [15,16]. More recently, an increasing number of DNN-based approaches integrated multi-omics data and systems biology structures into the learned models. Such approaches aim to adopt the DNN model on prior biological and medical knowledge and thereby improve our understanding of diseases and the effect mechanisms of medication in a new way.

In this review, we will outline the paradigms of NGS-based cancer research during the last decades. After that, we will discuss state-of-the-art systems biology, machine learning, and DNN-based approaches applied in various cancer studies. At first, we will summarize various challenges raised by the analysis of NGS datasets and cancer genomics, and we will discuss different applications of systems biology in cancer research. Additionally, we will introduce network-oriented machine learning methodologies that have been applied to NGS cancer data. Finally, we will discuss very recent DNN-based approaches in cancer researches.

### 1.1. Molecular Characterization and Multi-Omics Data

During the past decade, the field of cancer genomics expanded enormously, triggered by the accumulation of massive amounts of various omics data originating from different cancer tissue types stored in large public databases such as TCGA [17]. TCGA is a large-scale data generation project, which tremendously nourished research in cancer systems biology and cancer genomics. In particular, large-scale multi-omics data facilitate obtaining a molecular landscape of tumor development at different levels of multi-omics layers. Among various omics studies, genomics often focused on identifying driver mutations. These studies marked the starting point to gain a deeper understanding of cancer characteristics and development. Recently, a pan-cancer analysis of whole genomes (PCAWG) was able to identify these genomic features across the 38 different cancer types [18]. In addition, several target drug studies followed this cancer genomics approach in order to improve treatment of patients [19,20]. In recent years, the described discoveries paved the way for clinical applications of NGS. For instance, gene expression profiles are considered in the treatment plans of cancer patients. The most famous example application is breast cancer subtyping by PAM50. PAM50 is a gene panel tailored for molecular subtyping of breast tumors, based on a microarray subtype analysis. PAM50 uses relative expression levels of 50 genes for breast cancer subtype classifications, and it has shown great success in diagnosis and treatment strategies for breast cancer [21,22]. After this success, many studies aimed at developing valid tumor-subtyping panels or prognosis models for different tumor types. The development of more advanced NGS technology accelerated the research for treatment-specific cancer subtypes and corresponding panels [23]. In contrast, the study of epigenomics in cancer focuses on unique regulatory mechanisms altering molecular pathways. Such epigenomic alterations can cause abnormal activity of signaling pathways and can be associated with tumor development [24].

The application of machine learning on omics data has become a state-of-the-art approach to classify tumor subtypes originating from various tissues. With the aid of these large-scale NGS datasets, molecular characterization and subtyping have been done in both intra-cancer types and across the different cancer types. The TCGA’s early pan-cancer study focuses on molecular subtyping across the border of originating tissues. They were able to classify tumors with new molecular characteristics observed through NGS technology. For example, bladder cancer can be divided into two different subtypes, and one of them is more closely related to squamous cell carcinoma in the lung and head and neck cancer types [4]. Moreover, methods aiming to understand the interactions between genomics, transcriptomics, or epigenomic variants were introduced. These multi-platform and pan-cancer tumor studies have the potential to provide a more comprehensive understanding of the molecular mechanisms in tumors [25].

### 1.2. Tumor Heterogeneity: Cancer Genomics to Translational Medicine

As described in the previous section, the combination of high-throughput sequencing technology and bioinformatics analyses extended our knowledge of cancer’s molecular background. This data-driven analysis supported the use of these approaches for molecular diagnosis and genomic medicine [26,27]. Subsequently, the NGS research utilized omics data in various cancer researches such as molecular subtyping, prognosis analysis, and drug target identifications [3,23]. However, the attempt to connect in silico data-driven analysis to in vivo translational medicine highlighted several issues concerning robustness and reproducibility [28].

There might be various reasons for those issues, such as tumor microenvironment and tumor heterogeneity [29,30]. Every tumor has unique characteristics leading not only to inter-patient heterogeneity but also to intra-patient and intra-tumor heterogeneity. A single tumor comprises various clones defined by their unique mutational signatures, and thus a metastatic tumor can have different characteristics and different responses to certain drugs [31,32,33]. Although the detailed mechanisms between tumor cells and other normal cells are unclear, many studies reveal that the complex interactions between tumor and immune cells affect tumor growth and prognosis [34,35]. Due to the fact that bulk-cell sequencing can capture various cell types in a sample in a single shot, NGS cancer research has addressed two challenges, namely, clonal heterogeneity and infiltration of immune-related cells. Therefore, novel sequencing technology and new computational methodologies are developed to investigate the landscape of tumor microenvironments in various omics levels [36]. For instance, transcriptome data was analyzed to observe the landscape of immune infiltration [37]. A recent large cohort study of lymphoma is showing the importance of a tumor microenvironment and its clinical importance. They identified novel microenvironments subtypes by defining 25 functional gene expression signatures reflecting pathway activities from transcriptome data [38]. Genomic data was analyzed to identify cancer clones by their unique mutational signatures [39]. Moreover, with metastatic tumor or mouse xenograft model datasets, dynamics of clonal evolution were observable [40]. Clonal evolution and clonal expansion can explain tumor heterogeneity and its response to anticancer drugs. A mouse xenograft study revealed clonal dynamics, and it is a deterministic pattern in the model [41]. Furthermore, a recent study profiled the metastatic characteristics in each of 500 cancer cell lines. This study also indirectly supports deterministic clonal dynamics induced by the cell’s characters [42].

### 1.3. Drug Target Identification

NGS data have proven to be particularly useful to analyze and understand the mechanisms and effects of drugs. Thus, NGS applications have accelerated target drug discovery and development. In particular, the molecular characteristics of cancer were robustly translated into a potential treatment target such as cell-cycle pathway or PI3K pathway [20,43]. Furthermore, individual genomic alterations in cancer types that were found are a potential target for new drugs [44]. However, for some NGS-based studies a reproducibility issue was reported [28]. As depicted previously, one reason for this is tumor heterogeneity which makes cancer treatment more difficult [31,32]. A long-term follow-up NGS study identified the effect of cancer drugs in re-occurred or metastatic tumors by showing different molecular characteristics and clonal structures [45,46]. Additionally, another NGS-based study with a large cohort identified that genomic variants are abundant in the human population, i.e., drug effectiveness studies can be biased by populations and their different rare variants [47].

The following sections will address how bioinformatics algorithms and machine learning approaches have been used in the field to address the aforementioned issues.

## 2. Systems Biology in Cancer Research

Genes and their functions have been classified into gene sets based on experimental data. Our understandings of cancer concentrated into cancer hallmarks that define the characteristics of a tumor [48]. This collective knowledge is used for the functional analysis of unseen data. For instance, gene set enrichment analysis (GSEA) is a representative tool for systematic analysis using prior knowledge [49,50,51]. Furthermore, the regulatory relationships among genes were investigated, and, based on that, a pathway can be composed [1,52]. In this manner, the accumulation of public high-throughput sequencing data raised many big-data challenges and opened new opportunities and areas of application for computer science. Two of the most vibrantly evolving areas are systems biology and machine learning which tackle different tasks such as understanding the cancer pathways [9], finding crucial genes in pathways [22,53], or predicting functions of unidentified or understudied genes [54]. Essentially, those models include prior knowledge to develop an analysis and enhance interpretability for high-dimensional data [2]. In addition to understanding cancer pathways with in silico analysis, pathway activity analysis incorporating two different types of data, pathways and omics data, is developed to understand heterogeneous characteristics of the tumor and cancer molecular subtyping. Due to its advantage in interpretability, various pathway-oriented methods are introduced and become a useful tool to understand a complex diseases such as cancer [55,56,57].

In this section, we will discuss how two related research fields, namely, systems biology and machine learning, can be integrated with three different approaches (see Figure 2), namely, biological network analysis for biomarker validation, the use of machine learning with systems biology, and network-based models.

### 2.1. Biological Network Analysis for Biomarker Validation

The detection of potential biomarkers indicative of specific cancer types or subtypes is a frequent goal of NGS data analysis in cancer research. For instance, a variety of bioinformatics tools and machine learning models aim at identify lists of genes that are significantly altered on a genomic, transcriptomic, or epigenomic level in cancer cells. Typically, statistical and machine learning methods are employed to find an optimal set of biomarkers, such as single nucleotide polymorphisms (SNPs), mutations, or differentially expressed genes crucial in cancer progression. Traditionally, resource-intensive in vitro analysis was required to discover or validate those markers. Therefore, systems biology offers in silico solutions to validate such findings using biological pathways or gene ontology information (Figure 2b) [58]. Subsequently, gene set enrichment analysis (GSEA) [50] or gene set analysis (GSA) [59] can be used to evaluate whether these lists of genes are significantly associated with cancer types and their specific characteristics. GSA, for instance, is available via web services like DAVID [60] and g:Profiler [61]. Moreover, other applications use gene ontology directly [62,63]. In addition to gene-set-based analysis, there are other methods that focuse on the topology of biological networks. These approaches evaluate various network structure parameters and analyze the connectivity of two genes or the size and interconnection of their neighbors [64,65]. According to the underlying idea, the mutated gene will show dysfunction and can affect its neighboring genes. Thus, the goal is to find abnormalities in a specific set of genes linked with an edge in a biological network. For instance, KeyPathwayMiner can extract informative network modules in various omics data [66]. In summary, these approaches aim at predicting the effect of dysfunctional genes among neighbors according to their connectivity or distances from specific genes such as hubs [67,68]. During the past few decades, the focus of cancer systems biology extended towards the analysis of cancer-related pathways since those pathways tend to carry more information than a gene set. Such analysis is called Pathway Enrichment Analysis (PEA) [69,70]. The use of PEA incorporates the topology of biological networks. However, simultaneously, the lack of coverage issue in pathway data needs to be considered. Because pathway data does not cover all known genes yet, an integration analysis on omics data can significantly drop in genes when incorporated with pathways. Genes that can not be mapped to any pathway are called ‘pathway orphan.’ In this manner, Rahmati et al. introduced a possible solution to overcome the ‘pathway orphan’ issue [71]. At the bottom line, regardless of whether researchers consider gene-set or pathway-based enrichment analysis, the performance and accuracy of both methods are highly dependent on the quality of the external gene-set and pathway data [72].

### 2.2. De Novo Construction of Biological Networks

While the known fraction of existing biological networks barely scratches the surface of the whole system of mechanisms occurring in each organism, machine learning models can improve on known network structures and can guide potential new findings [73,74]. This area of research is called de novo network construction (Figure 2c), and its predictive models can accelerate experimental validation by lowering time costs [75,76]. This interplay between in silico biological networks building and mining contributes to expanding our knowledge in a biological system. For instance, a gene co-expression network helps discover gene modules having similar functions [77]. Because gene co-expression networks are based on expressional changes under specific conditions, commonly, inferring a co-expression network requires many samples. The WGCNA package implements a representative model using weighted correlation for network construction that leads the development of the network biology field [78]. Due to NGS developments, the analysis of gene co-expression networks subsequently moved from microarray-based to RNA-seq based experimental data [79]. However, integration of these two types of data remains tricky. Ballouz et al. compared microarray and NGS-based co-expression networks and found the existence of a bias originating from batch effects between the two technologies [80]. Nevertheless, such approaches are suited to find disease-specific co-expressional gene modules. Thus, various studies based on the TCGA cancer co-expression network discovered characteristics of prognostic genes in the network [81]. Accordingly, a gene co-expression network is a condition-specific network rather than a general network for an organism. Gene regulatory networks can be inferred from the gene co-expression network when various data from different conditions in the same organism are available. Additionally, with various NGS applications, we can obtain multi-modal datasets about regulatory elements and their effects, such as epigenomic mechanisms on transcription and chromatin structure. Consequently, a gene regulatory network can consist of solely protein-coding genes or different regulatory node types such as transcription factors, inhibitors, promoter interactions, DNA methylations, and histone modifications affecting the gene expression system [82,83]. More recently, researchers were able to build networks based on a particular experimental setup. For instance, functional genomics or CRISPR technology enables the high-resolution regulatory networks in an organism [84]. Other than gene co-expression or regulatory networks, drug target, and drug repurposing studies are active research areas focusing on the de novo construction of drug-to-target networks to allow the potential repurposing of drugs [76,85].

### 2.3. Network Based Machine Learning

A network-based machine learning model directly integrates the insights of biological networks within the algorithm (Figure 2d) to ultimately improve predictive performance concerning cancer subtyping or susceptibility to therapy. Following the establishment of high-quality biological networks based on NGS technologies, these biological networks were suited to be integrated into advanced predictive models. In this manner, Zhang et al., categorized network-based machine learning approaches upon their usage into three groups: (i) model-based integration, (ii) pre-processing integration, and (iii) post-analysis integration [7]. Network-based models map the omics data onto a biological network, and proper algorithms travel the network while considering both values of nodes and edges and network topology. In the pre-processing integration, pathway or other network information is commonly processed based on its topological importance. Meanwhile, in the post-analysis integration, omics data is processed solely before integration with a network. Subsequently, omics data and networks are merged and interpreted. The network-based model has advantages in multi-omics integrative analysis. Due to the different sensitivity and coverage of various omics data types, a multi-omics integrative analysis is challenging. However, focusing on gene-level or protein-level information enables a straightforward integration [86,87]. Consequently, when different machine learning approaches tried to integrate two or more different data types to find novel biological insights, one of the solutions is reducing the search space to gene or protein level and integrated heterogeneous datatypes [25,88].

In summary, using network information opens new possibilities for interpretation. However, as mentioned earlier, several challenges remain, such as the coverage issue. Current databases for biological networks do not cover the entire set of genes, transcripts, and interactions. Therefore, the use of networks can lead to loss of information for gene or transcript orphans. The following section will focus on network-based machine learning models and their application in cancer genomics. We will put network-based machine learning into the perspective of the three main areas of application, namely, molecular characterization, tumor heterogeneity analysis, and cancer drug discovery.

## 3. Network-Based Learning in Cancer Research

As introduced previously, the integration of machine learning with the insights of biological networks (Figure 2d) ultimately aims at improving predictive performance and interpretability concerning cancer subtyping or treatment susceptibility.

### 3.1. Molecular Characterization with Network Information

Various network-based algorithms are used in genomics and focus on quantifying the impact of genomic alteration. By employing prior knowledge in biological network algorithms, performance compared to non-network models can be improved. A prominent example is HotNet. The algorithm uses a thermodynamics model on a biological network and identifies driver genes, or prognostic genes, in pan-cancer data [89]. Another study introduced a network-based stratification method to integrate somatic alterations and expression signatures with network information [90]. These approaches use network topology and network-propagation-like algorithms. Network propagation presumes that genomic alterations can affect the function of neighboring genes. Two genes will show an exclusive pattern if two genes complement each other, and the function carried by those two genes is essential to an organism [91]. This unique exclusive pattern among genomic alteration is further investigated in cancer-related pathways. Recently, Ku et al. developed network-centric approaches and tackled robustness issues while studying synthetic lethality [92]. Although synthetic lethality was initially discovered in model organisms of genetics, it helps us to understand cancer-specific mutations and their functions in tumor characteristics [91].

Furthermore, in transcriptome research, network information is used to measure pathway activity and its application in cancer subtyping. For instance, when comparing the data of two or more conditions such as cancer types, GSEA as introduced in Section 2 is a useful approach to get an overview of systematic changes [50]. It is typically used at the beginning of a data evaluation [93]. An experimentally validated gene set can provide information about how different conditions affect molecular systems in an organism. In addition to the gene sets, different approaches integrate complex interaction information into GSEA and build network-based models [70]. In contrast to GSEA, pathway activity analysis considers transcriptome data and other omics data and structural information of a biological network. For example, PARADIGM uses pathway topology and integrates various omics in the analysis to infer a patient-specific status of pathways [94]. A benchmark study with pan-cancer data recently reveals that using network structure can show better performance [57]. In conclusion, while the loss of data is due to the incompleteness of biological networks, their integration improved performance and increased interpretability in many cases.

### 3.2. Tumor Heterogeneity Study with Network Information

The tumor heterogeneity can originate from two directions, clonal heterogeneity and tumor impurity. Clonal heterogeneity covers genomic alterations within the tumor [95]. While de novo mutations accumulate, the tumor obtains genomic alterations with an exclusive pattern. When these genomic alterations are projected on the pathway, it is possible to observe exclusive relationships among disease-related genes. For instance, the CoMEt and MEMo algorithms examine mutual exclusivity on protein–protein interaction networks [96,97]. Moreover, the relationship between genes can be essential for an organism. Therefore, models analyzing such alterations integrate network-based analysis [98].

In contrast, tumor purity is dependent on the tumor microenvironment, including immune-cell infiltration and stromal cells [99]. In tumor microenvironment studies, network-based models are applied, for instance, to find immune-related gene modules. Although the importance of the interaction between tumors and immune cells is well known, detailed mechanisms are still unclear. Thus, many recent NGS studies employ network-based models to investigate the underlying mechanism in tumor and immune reactions. For example, McGrail et al. identified a relationship between the DNA damage response protein and immune cell infiltration in cancer. The analysis is based on curated interaction pairs in a protein–protein interaction network [100]. Most recently, Darzi et al. discovered a prognostic gene module related to immune cell infiltration by using network-centric approaches [101]. Tu et al. presented a network-centric model for mining subnetworks of genes other than immune cell infiltration by considering tumor purity [102].

### 3.3. Drug Target Identification with Network Information

In drug target studies, network biology is integrated into pharmacology [103]. For instance, Yamanishi et al. developed novel computational methods to investigate the pharmacological space by integrating a drug-target protein network with genomics and chemical information. The proposed approaches investigated such drug-target network information to identify potential novel drug targets [104]. Since then, the field has continued to develop methods to study drug target and drug response integrating networks with chemical and multi-omic datasets. In a recent survey study by Chen et al., the authors compared 13 computational methods for drug response prediction. It turned out that gene expression profiles are crucial information for drug response prediction [105].

Moreover, drug-target studies are often extended to drug-repurposing studies. In cancer research, drug-repurposing studies aim to find novel interactions between non-cancer drugs and molecular features in cancer. Drug-repurposing (or repositioning) studies apply computational approaches and pathway-based models and aim at discovering potential new cancer drugs with a higher probability than de novo drug design [16,106]. Specifically, drug-repurposing studies can consider various areas of cancer research, such as tumor heterogeneity and synthetic lethality. As an example, Lee et al. found clinically relevant synthetic lethality interactions by integrating multiple screening NGS datasets [107]. This synthetic lethality and related-drug datasets can be integrated for an effective combination of anticancer therapeutic strategy with non-cancer drug repurposing.

## 4. Deep Learning in Cancer Research

DNN models develop rapidly and become more sophisticated. They have been frequently used in all areas of biomedical research. Initially, its development was facilitated by large-scale imaging and video data. While most data sets in the biomedical field would not typically be considered big data, the rapid data accumulation enabled by NGS made it suitable for the application of DNN models requiring a large amount of training data [108]. For instance, in 2019, Samiei et al. used TCGA-based large-scale cancer data as benchmark datasets for bioinformatics machine learning research such as Image-Net in the computer vision field [109]. Subsequently, large-scale public cancer data sets such as TCGA encouraged the wide usage of DNNs in the cancer domain [110]. Over the last decade, these state-of-the-art machine learning methods have been incorporated in many different biological questions [111].

In addition to public cancer databases such as TCGA, the genetic information of normal tissues is stored in well-curated databases such as GTEx [112] and 1000Genomes [113]. These databases are frequently used as control or baseline training data for deep learning [114]. Moreover, other non-curated large-scale data sources such as GEO (https://www.ncbi.nlm.nih.gov/geo/, accessed on 20 May 2021) can be leveraged to tackle critical aspects in cancer research. They store a large-scale of biological data produced under various experimental setups (Figure 1). Therefore, an integration of GEO data and other data requires careful preprocessing. Overall, an increasing amount of datasets facilitate the development of current deep learning in bioinformatics research [115].

### 4.1. Challenges for Deep Learning in Cancer Research

Many studies in biology and medicine used NGS and produced large amounts of data during the past few decades, moving the field to the big data era. Nevertheless, researchers still face a lack of data in particular when investigating rare diseases or disease states. Researchers have developed a manifold of potential solutions to overcome this lack of data challenges, such as imputation, augmentation, and transfer learning (Figure 3b). Data imputation aims at handling data sets with missing values [116]. It has been studied on various NGS omics data types to recover missing information [117]. It is known that gene expression levels can be altered by different regulatory elements, such as DNA-binding proteins, epigenomic modifications, and post-transcriptional modifications. Therefore, various models integrating such regulatory schemes have been introduced to impute missing omics data [118,119]. Some DNN-based models aim to predict gene expression changes based on genomics or epigenomics alteration. For instance, TDimpute aims at generating missing RNA-seq data by training a DNN on methylation data. They used TCGA and TARGET (https://ocg.cancer.gov/programs/target/data-matrix, accessed on 20 May 2021) data as proof of concept of the applicability of DNN for data imputation in a multi-omics integration study [120]. Because this integrative model can exploit information in different levels of regulatory mechanisms, it can build a more detailed model and achieve better performance than a model build on a single-omics dataset [117,121]. The generative adversarial network (GAN) is a DNN structure for generating simulated data that is different from the original data but shows the same characteristics [122]. GANs can impute missing omics data from other multi-omics sources. Recently, the GAN algorithm is getting more attention in single-cell transcriptomics because it has been recognized as a complementary technique to overcome the limitation of scRNA-seq [123]. In contrast to data imputation and generation, other machine learning approaches aim to cope with a limited dataset in different ways. Transfer learning or few-shot learning, for instance, aims to reduce the search space with similar but unrelated datasets and guide the model to solve a specific set of problems [124]. These approaches train models with data of similar characteristics and types but different data to the problem set. After pre-training the model, it can be fine-tuned with the dataset of interest [125,126]. Thus, researchers are trying to introduce few-shot learning models and meta-learning approaches to omics and translational medicine. For example, Select-ProtoNet applied the ProtoTypical Network [127] model to TCGA transcriptome data and classified patients into two groups according to their clinical status [128]. AffinityNet predicts kidney and uterus cancer subtypes with gene expression profiles [129].

### 4.2. Molecular Charactization with Network and DNN Model

DNNs have been applied in multiple areas of cancer research. For instance, a DNN model trained on TCGA cancer data can aid molecular characterization by identifying cancer driver genes. At the very early stage, Yuan et al. build DeepGene, a cancer-type classifier. They implemented data sparsity reduction methods and trained the DNN model with somatic point mutations [130]. Lyu et al. [131] and DeepGx [132] embedded a 1-D gene expression profile to a 2-D array by chromosome order to implement the convolution layer (Figure 3a). Other algorithms, such as the deepDriver, use k-nearest neighbors for the convolution layer. A predefined number of neighboring gene mutation profiles was the input for the convolution layer. It employed this convolution layer in a DNN by aggregating mutation information of the k-nearest neighboring genes [11]. Instead of embedding to a 2-D image, DeepCC transformed gene expression data into functional spectra. The resulting model was able to capture molecular characteristics by training cancer subtypes [14].

Another DNN model was trained to infer the origin of tissue from single-nucleotide variant (SNV) information of metastatic tumor. The authors built a model by using the TCGA/ICGC data and analyzed SNV patterns and corresponding pathways to predict the origin of cancer. They discovered that metastatic tumors retained their original cancer’s signature mutation pattern. In this context, their DNN model obtained even better accuracy than a random forest model [133] and, even more important, better accuracy than human pathologists [12].

### 4.3. Tumor Heterogeneity with Network and DNN Model

As described in Section 4.1, there are several issues because of cancer heterogeneity, e.g., tumor microenvironment. Thus, there are only a few applications of DNN in intratumoral heterogeneity research. For instance, Menden et al. developed ’Scaden’ to deconvolve cell types in bulk-cell sequencing data. ’Scaden’ is a DNN model for the investigation of intratumor heterogeneity. To overcome the lack of training datasets, researchers need to generate in silico simulated bulk-cell sequencing data based on single-cell sequencing data [134]. It is presumed that deconvolving cell types can be achieved by knowing all possible expressional profiles of the cell [36]. However, this information is typically not available. Recently, to tackle this problem, single-cell sequencing-based studies were conducted. Because of technical limitations, we need to handle lots of missing data, noises, and batch effects in single-cell sequencing data [135]. Thus, various machine learning methods were developed to process single-cell sequencing data. They aim at mapping single-cell data onto the latent space. For example, scDeepCluster implemented an autoencoder and trained it on gene-expression levels from single-cell sequencing. During the training phase, the encoder and decoder work as denoiser. At the same time, they can embed high-dimensional gene-expression profiles to lower-dimensional vectors [136]. This autoencoder-based method can produce biologically meaningful feature vectors in various contexts, from tissue cell types [137] to different cancer types [138,139].

### 4.4. Drug Target Identification with Networks and DNN Models

In addition to NGS datasets, large-scale anticancer drug assays enabled the training train of DNNs. Moreover, non-cancer drug response assay datasets can also be incorporated with cancer genomic data. In cancer research, a multidisciplinary approach was widely applied for repurposing non-oncology drugs to cancer treatment. This drug repurposing is faster than de novo drug discovery. Furthermore, combination therapy with a non-oncology drug can be beneficial to overcome the heterogeneous properties of tumors [85]. The deepDR algorithm integrated ten drug-related networks and trained deep autoencoders. It used a random-walk-based algorithm to represent graph information into feature vectors. This approach integrated network analysis with a DNN model validated with an independent drug-disease dataset [15].

The authors of CDRscan did an integrative analysis of cell-line-based assay datasets and other drug and genomics datasets. It shows that DNN models can enhance the computational model for improved drug sensitivity predictions [140]. Additionally, similar to previous network-based models, the multi-omics application of drug-targeted DNN studies can show higher prediction accuracy than the single-omics method. MOLI integrated genomic data and transcriptomic data to predict the drug responses of TCGA patients [141].

### 4.5. Graph Neural Network Model

In general, the advantage of using a biological network is that it can produce more comprehensive and interpretable results from high-dimensional omics data. Furthermore, in an integrative multi-omics data analysis, network-based integration can improve interpretability over traditional approaches. Instead of pre-/post-integration of a network, recently developed graph neural networks use biological networks as the base structure for the learning network itself. For instance, various pathways or interactome information can be integrated as a learning structure of a DNN and can be aggregated as heterogeneous information. In a GNN study, a convolution process can be done on the provided network structure of data. Therefore, the convolution on a biological network made it possible for the GNN to focus on the relationship among neighbor genes. In the graph convolution layer, the convolution process integrates information of neighbor genes and learns topological information (Figure 3d). Consequently, this model can aggregate information from far-distant neighbors, and thus can outperform other machine learning models [142].

In the context of the inference problem of gene expression, the main question is whether the gene expression level can be explained by aggregating the neighboring genes. A single gene inference study by Dutil et al. showed that the GNN model outperformed other DNN models [143]. Moreover, in cancer research, such GNN models can identify cancer-related genes with better performance than other network-based models, such as HotNet2 and MutSigCV [144]. A recent GNN study with a multi-omics integrative analysis identified 165 new cancer genes as an interactive partner for known cancer genes [145]. Additionally, in the synthetic lethality area, dual-dropout GNN outperformed previous bioinformatics tools for predicting synthetic lethality in tumors [146]. GNNs were also able to classify cancer subtypes based on pathway activity measures with RNA-seq data. Lee et al. implemented a GNN for cancer subtyping and tested five cancer types. Thus, the informative pathway was selected and used for subtype classification [147]. Furthermore, GNNs are also getting more attention in drug repositioning studies. As described in Section 3.3, drug discovery requires integrating various networks in both chemical and genomic spaces (Figure 3d). Chemical structures, protein structures, pathways, and other multi-omics data were used in drug-target identification and repurposing studies (Figure 3c). Each of the proposed applications has a specialty in the different purposes of drug-related tasks. Sun et al. summarized GNN-based drug discovery studies and categorized them into four classes: molecular property and activity prediction, interaction prediction, synthesis prediction, and de novo drug design. The authors also point out four challenges in the GNN-mediated drug discovery. At first, as we described before, there is a lack of drug-related datasets. Secondly, the current GNN models can not fully represent 3-D structures of chemical molecules and protein structures. The third challenge is integrating heterogeneous network information. Drug discovery usually requires a multi-modal integrative analysis with various networks, and GNNs can improve this integrative analysis. Lastly, although GNNs use graphs, stacked layers still make it hard to interpret the model [148].

### 4.6. Shortcomings in AI and Revisiting Validity of Biological Networks as Prior Knowledge

The previous sections reviewed a variety of DNN-based approaches that present a good performance on numerous applications. However, it is hardly a panacea for all research questions. In the following, we will discuss potential limitations of the DNN models. In general, DNN models with NGS data have two significant issues: (i) data requirements and (ii) interpretability. Usually, deep learning needs a large proportion of training data for reasonable performance which is more difficult to achieve in biomedical omics data compared to, for instance, image data. Today, there are not many NGS datasets that are well-curated and -annotated for deep learning. This can be an answer to the question of why most DNN studies are in cancer research [110,149]. Moreover, the deep learning models are hard to interpret and are typically considered as black-boxes. Highly stacked layers in the deep learning model make it hard to interpret its decision-making rationale. Although the methodology to understand and interpret deep learning models has been improved, the ambiguity in the DNN models’ decision-making hindered the transition between the deep learning model and translational medicine [149,150].

As described before, biological networks are employed in various computational analyses for cancer research. The studies applying DNNs demonstrated many different approaches to use prior knowledge for systematic analyses. Before discussing GNN application, the validity of biological networks in a DNN model needs to be shown. The LINCS program analyzed data of ’The Connectivity Map (CMap) project’ to understand the regulatory mechanism in gene expression by inferring the whole gene expression profiles from a small set of genes (https://lincsproject.org/, accessed on 20 May 2021) [151,152]. This LINCS program found that the gene expression level is inferrable with only nearly 1000 genes. They called this gene list ’landmark genes’. Subsequently, Chen et al. started with these 978 landmark genes and tried to predict other gene expression levels with DNN models. Integrating public large-scale NGS data showed better performance than the linear regression model. The authors conclude that the performance advantage originates from the DNN’s ability to model non-linear relationships between genes [153].

Following this study, Beltin et al. extensively investigated various biological networks in the same context of the inference of gene expression level. They set up a simplified representation of gene expression status and tried to solve a binary classification task. To show the relevance of a biological network, they compared various gene expression levels inferred from a different set of genes, neighboring genes in PPI, random genes, and all genes. However, in the study incorporating TCGA and GTEx datasets, the random network model outperformed the model build on a known biological network, such as StringDB [154]. While network-based approaches can add valuable insights to analysis, this study shows that it cannot be seen as the panacea, and a careful evaluation is required for each data set and task. In particular, this result may not represent biological complexity because of the oversimplified problem setup, which did not consider the relative gene-expressional changes. Additionally, the incorporated biological networks may not be suitable for inferring gene expression profiles because they consist of expression-regulating interactions, non-expression-regulating interactions, and various in vivo and in vitro interactions.

## 5. Conclusions

The advance of NGS and subsequent massive production of NGS data facilitated various NGS analysis pipelines. The interplay between NGS and bioinformatics widened the spectrum of applicability of NGS and enabled the extraction and uncovering of detailed insights into the molecular mechanisms of cancer development and treatment. Accordingly, the tailored combination with machine learning and systems biology broadens our knowledge from genetics to medicine, particularly in cancer research. This review shows the development of NGS and corresponding computational methods in cancer research over the past few decades. We mainly focus on the following three levels, machine learning, machine learning with systems biology, and more recent approaches in deep learning. Among the vast amount of research trends in NGS applications, we focused on NGS data interpretations and how this developed methodology improved our understanding of cancer. In cancer genomics, sequencing data analysis extended from finding single mutated genes to mining cancer driver gene sets and their effect on biological pathways. With pathway information, genomics research was able to extend to synthetic lethality and drug combination. Similar to genomics, the cancer transcriptomics field moved from searching for differentially expressed genes to pathway activity measurement and thereby contributed to the development of a diagnosis panel. In systems biology, transcriptomics is widely used for de novo network reconstruction and finding novel pathways. Moreover, recently, deep learning algorithms are applied and were able to outperform existing models on various tasks, showing the power of a new data-driven methodology. This shift was observable in different areas of cancer research, biomarker identification, molecular characterization of cancer, tumor heterogeneity, and target-drug repurposing. However, although recently sophisticated applications of deep learning showed improved accuracy, it does not reflect a general advancement. Depending on the type of NGS data, the experimental design, and the question to be answered, a proper approach and specific deep learning algorithms need to be considered. Deep learning is not a panacea. In general, to employ machine learning and systems biology methodology for a specific type of NGS data, a certain experimental design, a particular research question, the technology, and network data have to be chosen carefully.

In the last few decades, NGS enabled an improvement of diagnostics in medicine from genetic diseases to cancer. Many recent machine learning-based studies report their advancements in disease classification and prognosis analysis. However, because machine learning models are often treated as black boxes and lack transparency, the lack of explainability leads to poor acceptance among clinicians. Especially for DNN-based models, it is often hard to understand why it shows better accuracy than other methods. To solve this ’Black box’ issue, the novel field of “explainable AI” aims at improving visualization, explanation, and interpretation of machine learning models such as deep learning. Explainable AI aims to fill the interpretability gap between translational medicine and machine learning. Furthermore, it can be beneficial to improve on machine learning methods itself by identifying their weakness. For instance, interactive machine learning or human-in-the-loop AI, where machine learning models are updated based on a human expert, can be a solution to both an interpretability issue and an improving machine learning model [155,156]. In the work by Augusto et al., sequential rule mining was proposed as one of the solutions to understand the ’black-box’ machine learning model. Their pipeline was able to find biologically relevant genes in six different datasets [157]. Other approaches include the calibration of machine learning models by making the predictions probabilistically interpretable [158]. Another way to improve interpretability is the use of graph-based models, as introduced in Section 4.5. As mentioned earlier, GNNs have advantages in multi-omics integrated analysis and intrinsically allow for more explainability [159]. Various recent studies are reporting the merit of graph-oriented models. However, as described in Section 4.5, depending on the detailed structure of a problem, applying a graphical model may or may not lead to better performance. Here, a well-informed choice of biological network information and algorithms is crucial for the success of each specific analysis.

## Figures and Tables

**Figure 1 cancers-13-03148-f001:**
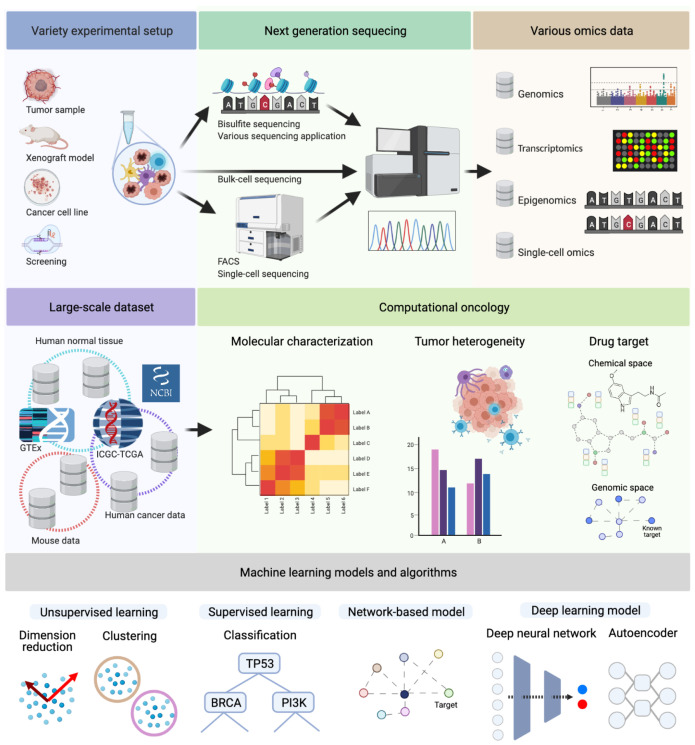
Next-generation sequencing data can originate from various experimental and technological conditions. Depending on the purpose of the experiment, one or more of the depicted omics types (Genomics, Transcriptomics, Epigenomics, or Single-Cell Omics) are analyzed. These approaches led to an accumulation of large-scale NGS datasets to solve various challenges of cancer research, molecular characterization, tumor heterogeneity, and drug target discovery. For instance, The Cancer Genome Atlas (TCGA) dataset contains multi-omics data from ten-thousands of patients. This dataset facilitates a variety of cancer researches for decades. Additionally, there are also independent tumor datasets, and, frequently, they are analyzed and compared with the TCGA dataset. As the large scale of omics data accumulated, various machine learning techniques are applied, e.g., graph algorithms and deep neural networks, for dimensionality reduction, clustering, or classification. (Created with BioRender.com.)

**Figure 2 cancers-13-03148-f002:**
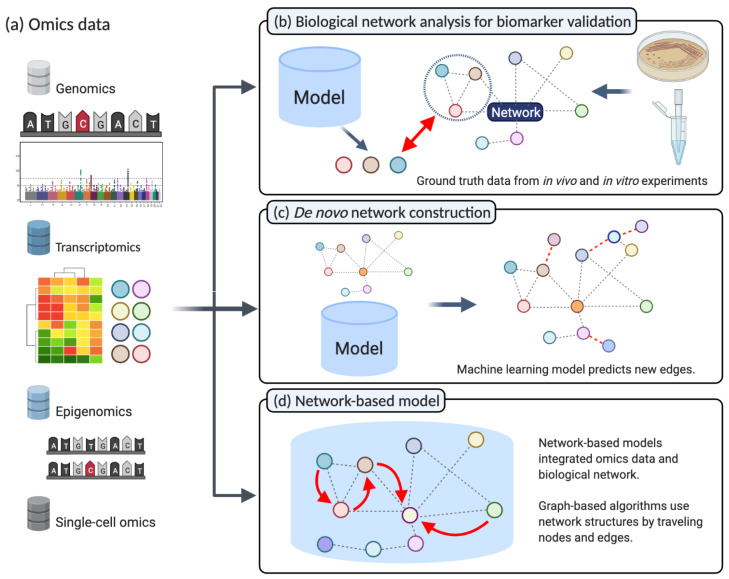
(**a**) A multitude of different types of data is produced by next-generation sequencing, for instance, in the fields of genomics, transcriptomics, and epigenomics. (**b**) Biological networks for biomarker validation: The in vivo or in vitro experiment results are considered ground truth. Statistical analysis on next-generation sequencing data produces candidate genes. Biological networks can validate these candidate genes and highlight the underlying biological mechanisms (Section 2.1). (**c**) De novo construction of Biological Networks: Machine learning models that aim to reconstruct biological networks can incorporate prior knowledge from different omics data. Subsequently, the model will predict new unknown interactions based on new omics information (Section 2.2). (**d**) Network-based machine learning: Machine learning models integrating biological networks as prior knowledge to improve predictive performance when applied to different NGS data (Section 2.3). (Created with BioRender.com).

**Figure 3 cancers-13-03148-f003:**
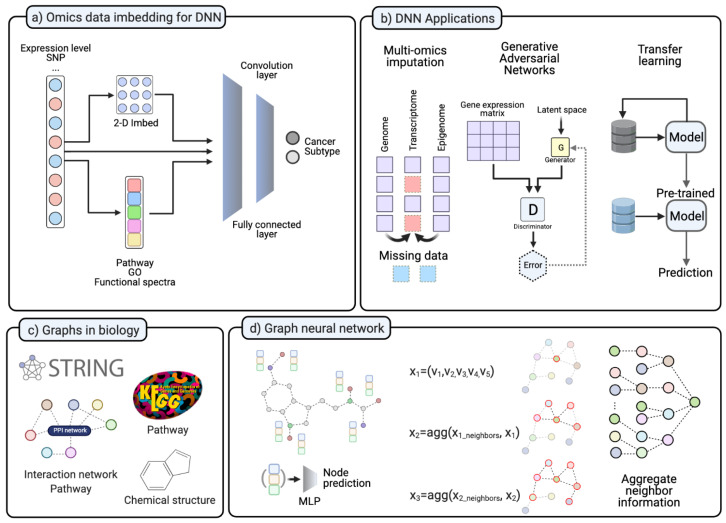
(**a**) In various studies, NGS data transformed into different forms. The 2-D transformed form is for the convolution layer. Omics data is transformed into pathway level, GO enrichment score, or Functional spectra. (**b**) DNN application on different ways to handle lack of data. Imputation for missing data in multi-omics datasets. GAN for data imputation and in silico data simulation. Transfer learning pre-trained the model with other datasets and fine-tune. (**c**) Various types of information in biology. (**d**) Graph neural network examples. GCN is applied to aggregate neighbor information. (Created with BioRender.com).

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
