# Peer review of "Integrative Analysis of Next-Generation Sequencing for Next-Generation Cancer Research toward Artificial Intelligence"

_cancers, 2021, doi:10.3390/cancers13133148_

Round 1
Reviewer 1 Report
The authors have organized the review well and the topic is current and of great interest. please correct the title of figure 1 (varaiety with variety).
Author Response
Reviewer 1: Comments and Suggestions for Authors
The authors have organized the review well and the topic is current and of great interest. please correct the title of figure 1 (varaiety with variety).
- We thank the reviewer for pointing out this mistake. We corrected the typo in Figure 1 (Varaiety -> Variety).
Reviewer 2 Report
The article is a concise review of applied AI . Although authors mention briefly AI limitation in conclusion the article needs a dedicated paragraph/section on AI shortcoming.
Author Response
Reviewer 2: Comments and Suggestions for Authors
The article is a concise review of applied AI . Although authors mention briefly AI limitation in conclusion the article needs a dedicated paragraph/section on AI shortcoming.
Reply:
We thank the reviewer for the insightful suggestion. We added a dedicated paragraph in section 4.6 and changed the section title to include a description of the general shortcoming in AI and Deep Learning. We point out well-known limitations in AI, requiring large-scale data and interpretability.
Reviewer 3 Report
The manuscript of Park and Coworkers is well-written and nicely illustrated. It represents a comprehensive review of the issue, whose reading needs some knowledge of bio-informatics. I have a few comments for the authors.
1) Since the artificial intelligence is not the primary object of the review, a slightly different title might better reflect its content (e.g. Next-Generation Sequencing Data Integration for Next-Generation Cancer Research: Moving Toward Artificial Intelligence).
2) In the context of paragraph 1.2, the recent publication of Kotlov N et al. (Cancer Discovery 2021, 11, 1468-1489) might be usefully quoted.
3) Analogously, in the context of paragraph 2.2 the contribution of Langfelder P and Horvath S (WGCNA: an R package for weighted correlation network analysis. BMC Bioinformatics 2008, 9 559) should be quoted and discussed.
Author Response
Reviewer 3: Comments and Suggestions for Authors
The manuscript of Park and Coworkers is well-written and nicely illustrated. It represents a comprehensive review of the issue, whose reading needs some knowledge of bio-informatics. I have a few comments for the authors.
1) Since the artificial intelligence is not the primary object of the review, a slightly different title might better reflect its content (e.g. Next-Generation Sequencing Data Integration for Next-Generation Cancer Research: Moving Toward Artificial Intelligence).
Reply: We thank the reviewer for the suggestion for the title. We changed the title to “Integrative analysis of Next-Generation Sequencing for Next-Generation Cancer Research toward Artificial Intelligence”
2) In the context of paragraph 1.2, the recent publication of Kotlov N et al. (Cancer Discovery 2021, 11, 1468-1489) might be usefully quoted.
Reply: We thank the reviewer for the suggestion. The study by Kotlov N et al. is a well-designed data integrative study. So it is well fitted in the tumor microenvironment section. We added the following sentences to section 1.2: “A recent large cohort study of lymphoma is showing the importance of a tumor microenvironment and its clinical importance. They identified novel microenvironments subtypes by defining 25 functional gene expression signatures reflecting pathway activities from transcriptome data.”
3) Analogously, in the context of paragraph 2.2 the contribution of Langfelder P and Horvath S (WGCNA: an R package for weighted correlation network analysis. BMC Bioinformatics 2008, 9 559) should be quoted and discussed.
Reply: We thank the reviewer for the suggestion. WGCNA is a landmark paper in de novo network construction study. We added the following sentence to section 2.2: “The WGCNA package implements a representative model using weighted correlation for network construction that leads the development of the network biology field.”